# A Perilipin Affects Lipid Droplet Homeostasis and Aerial Hyphal Growth, but Has Only Small Effects on Virulence in the Insect Pathogenic Fungus *Beauveria bassiana*

**DOI:** 10.3390/jof8060634

**Published:** 2022-06-15

**Authors:** Xiaoyun Wang, Yu Liu, Nemat O. Keyhani, Shengan Zhu, Jing Wang, Junyao Wang, Dan Jin, Yanhua Fan

**Affiliations:** 1Biotechnology Research Center, Southwest University, Beibei, Chongqing 400716, China; wangxy199312@outlook.com (X.W.); zsa19930303@email.swu.edu.cn (S.Z.); wangj199512@outlook.com (J.W.); w1992wjy@163.com (J.W.); jindancj@swu.edu.cn (D.J.); 2Key Laboratory of Sericultural Biology and Genetic Breeding, State Key Laboratory of Silkworm Genome Biology, Ministry of Agriculture and Rural Affairs, College of Sericulture, Textile and Biomass Sciences, Southwest University, Beibei, Chongqing 400716, China; lfiaun@swu.edu.cn; 3Department of Microbiology and Cell Science, University of Florida, Gainesville, FL 32611, USA; keyhani@ufl.edu

**Keywords:** entomopathogenic fungi, perilipin, lipid body, fungal growth

## Abstract

Lipid assimilation, storage, and turnover impact growth, development, and virulence in many microbial pathogens including fungi. Perilipins are proteins associated with lipid droplets (LDs) that mediate their assembly and turnover. Here, we characterized the *Beauveria bassiana* (*BbPlin1*) perilipin. *BbPlin1* expression was higher in minimal media than in rich media, and, using a BbPlin1::eGFP fusion protein, the protein was shown to be co–localized to LDs, with the high expression seen during infection and proliferation within the insect (*Galleria mellonella*) host that dramatically decreased to almost no expression during fungal outgrowth on cadavers including in conidia, but that *BbPlin1* production resumed in the conidia once placed in nutrient–containing media allowing for germination and growth. Characterization of a targeted gene deletion strain (Δ*BbPlin1*) revealed a dramatic (>30%) reduction in cellular LD content, promotion of aerial hyphal growth, and a small decrease in virulence, with little to no effects on vegetative growth and stress responses. However, in the Δ*BbPlin1* strain, expression of the complementary LD–associated caleosin gene, *BbCal1*, was enhanced under nutrient–poor conditions, although no changes in *BbPlin1* expression were seen in a Δ*BbCal1* strain and the expression of *BbPlin1* in the Δ*BbCal1* strain did not change LD patterns in cells. Transcriptome and RT–PCR analyses indicated increased expression of lipid metabolism–related genes, including triacylglyercol lipase 3, enoyl–CoA isomerase, and diacylglycerol–O–acetyl transferase in the *BbPlin1* deletion mutant. Lipid profile analyses confirmed that the loss of *BbPlin1* significantly reduced the cellular levels of contents of triacylglycerol, diacylglycerol, and phosphatidylethanolamine as compared to the wild–type strain. These results demonstrate the involvement of the *B. bassiana* perilipin in mediating lipid homeostasis, fungal aerial hyphal growth, and virulence, revealing critical cycling from high expression during nutrient utilization within host cadavers to low expression during growth on the surface of the cadaver during the infection process.

## 1. Introduction

Lipid droplets (LDs) are intracellular dynamic organelles that mainly consist of neutral lipids, including triglycerides (TGs) and steryl esters (SEs) [1,2]. LDs are surrounded by a phospholipid monolayer and associated proteins including caleosins. In addition to acting as storage reservoirs for lipids and maintaining lipid homeostasis, LDs have been shown to have significant broader physiological functions including mediating phospholipid recycling, acting to protect unfolded proteins, and contributing to toxin and even pathogen resistance [1,2]. Lipid storage disorders contribute to a range of metabolic diseases in human and mammalian animals [3]. In fungi, factors associated with LDs include structural proteins with both perilipin and caleosin proteins found at the LD membrane, as well as lipid metabolism–related enzymes (e.g., acetyl coenzyme A (CoA) carboxylase, acyl–CoA synthetase, diacylglycerol acyltransferase 2 (DGATE2), and lipases), and, depending upon the system, up to 200 protein components have been identified in LDs by proteome analysis, although many of these are also found in other organelles [2,4]. Perilipins that possess a conserved PAT (Perilipin/ADRP/TIP47) domain are one of the most abundant proteins found associated with LDs, and have been characterized in animal, plants, and fungi [5]. Perilipins coat the LD surfaces and restrict the access of cytosolic lipases to the lipids within the droplet [6]. In response to the appropriate stimulus, the perilipin protein, if phosphorylated by protein kinase A, changes protein partners and subsequently promotes lipolysis by recruiting lipases to the LD [7]. Five perilipin encoding genes with additional splice variants have been characterized in mammalian cells; however, fungi typically contain just a single perilipin gene [8,9]. Indeed, yeast does not have a perilipin, although an LDs–associated protein, Pet10p, has been identified that contains a PAT domain and appears to be functionally similar to perilipins. In the *Saccharomyces cerevisiase* Pet10p null mutant, both diacylglycerol acyltransferase (Dga1p) activity and TG accumulation were reduced by 30–50% [10]. Overall, characterization of fungal perilipins remains limited. Deletion of the CAP20 perilipin homolog in bitter rot fungus, *Colletotrichum gloesoporioodes*, led to the production of thinner conidia, smaller appressoria, and decreased virulence against rubber trees [11]. In the entomopathogenic fungus *Metarhizium robertsii*, deletion of the MPL1 perilipin gene resulted in thinner hyphae, reduced appressoria turgor pressure, and decreased virulence [12]. Similarly, in the fungal rice pathogen, *Magnaporthe oryzae*, the LDP1 perilipin was shown to be important for lipid mobilization during appressorial infection, and in the mycoparasitic fungus *Clonostachy rosea*, a perilipin was identified as highly expressed during infection of *Sclerotinia sclerotiorum* sclerotia [13,14]. Intriguingly, when the perilipin gene was over–expressed in the latter fungus (*C. rosea*), the parasitic ability of the fungus on host sclerotia was enhanced. However, although lipid assimilation is known to be important for the entomopathogenic fungus, *Beauveria bassiana* [15,16], it is unclear whether these fungi produce true appressoria, and the mechanisms by which they infect insects are significantly different from those of the related *Metarhizium* species [17,18,19].

*B. bassiana* is broad host range insect pathogen that has been commercialized for use as a pest control worldwide [20,21]. Infection is initiated by conidial attachment to the insect cuticle, followed by fungal cuticle penetration, hyphal growth reaching the hemocoel, and subsequent proliferation of yeast–like hyphal bodies in the hemolymph [19,22]. After host death, fungal hyphae grow out from the cadavers, producing aerial conidia that can cause a new round of infection under suitable conditions [23]. The ability to utilize (host and other) lipids has been shown to be not only linked to virulence but also stress response and the ability of the fungus to overcome host defenses [15,16,17,24,25]. Caleosins represent a protein family initially characterized as a major plant LD–associated protein that is also found in fungi, playing important roles in LD structural maintenance and turnover [26]. In a previous study, we identified a *B. bassiana* lipid droplet–associated protein homolog of caleosins, labeled *BbCal1*. Deletion of *BbCal1* resulted in changes in the cellular lipid profiles, led to the formation of unique vacuolar/endoplasmic reticulum/multilamellar vesicle–like structures, and impacted virulence, but had no effects on conidial germination, stress responses, or growth on lipid substrates [27,28]. In the present study, we characterized another lipid droplet–associated protein, namely the *B. bassiana* perilipin (BbPlin1). Expression profiles analyses and GFP–protein tagging revealed that this gene and its protein product exhibited high expression during growth on nutrient–poor media including Czapek–Dox broth and agar (CZB/CZA) and during growth on *B. bassiana*–killed cadavers. Deletion of *BbPlin1* significantly altered cellular lipid profiles, promoted the formation of aerial hyphal growth, but decreased fungal virulence against *Galleria mellonella* larvae.

## 2. Materials and Methods

### 2.1. Strains and Insect Larvae

*B. bassiana* Bb0062 (CGMCC 7.34) was used in this study and was routinely grown in Czapek–Dox broth/agar (CZB)/CZA, potato dextrose broth/agar (PDB)/PDA, and Sabouraud dextrose broth/agar (SDB)/SDA as indicated. *Escherichia coli* DH5α and *Agrobacterium tumefaciens* AGL–1 were used for DNA manipulations and fungal transformation, respectively. Greater wax moth, *Galleria mellonella* larvae, were used for insect bioassays (Keyun Biological Co., Jiyuan, China).

### 2.2. Molecular Manipulations

All primers used in this study are listed in Appendix A. Targeted gene knockout vectors for *BbPlin1* in *B. bassiana* were constructed via homologous recombination. Left/right gene sequences for *BbPlin1* were amplified by PCR using indicated primer pairs, with *Eco*RI/*Spe*I and *Xba*I/*Hind* III restriction sites designed into the primers, respectively. The obtained PCR fragments were digested with indicated restriction enzymes and cloned into the corresponding sites found on the pK2 vector that also contained the chlorimuron ethyl resistance gene marker (sur). To obtain the complementation vector, the entire *BbPlin1* gene including promoter (1972 kb) and gene (633 kb) sequences was amplified using genomic DNA as a template and cloned into pK2–Bar containing the phosphinothricin resistance gene marker (*bar*). To examine the expression and localization pattern of *BbPlin1*, the fusion construct *PBbPlin1::BbPlin1::eGFP* (carboxy–terminal protein fusion) was generated. Briefly, a *BbPlin1* fragment containing its native promoter was amplified by PCR and fused with the *eGFP* gene. The fusion construct was cloned into the pK2–sur vector via a unique *Spe*I site. All vector constructs were verified by restriction enzyme digestion and sequencing and then transformed into *A. tumefaciens* AGL–1. Fungal transformation mediated by *A. tumefaciens* was performed as described [29].

### 2.3. Expression Analysis

Gene expression analyses of *BbPlin1* were performed using real–time PCR and protein expression via fluorescent observations of the *PBbPlin1::eGFP* and *PBbPlin1::BbPlin1::eGFP* strains. For RT–PCR analysis, the *B. bassiana* wild–type strain was grown in liquid media (CZB, PDB, and SDB) for 3 days or on solid media (CZA, PDA, and SDA) for 7 days. Fungal cells were collected and the total RNA was extracted using the EASY Spin plant RNA rapid extraction kit of (Biomed Biotechnology Co., Shijingshan, Beijng, China) following the manufacturer’s instructions. cDNA was generated using the iScript II Reverse Transcriptase kit of Vazyme Biotech Co., Ltd. (Nanjing, China) and used for the Real–time PCR experiment performed using indicated primers (Appendix A) as follows: denaturation at 95 °C for 10 min, followed by 40 cycles of 30 s at 95 °C, 30 s at 56 °C, and 30 s at 72 °C. Relative expression levels were calculated by using Bio–Rad CFX Manager 3.0 software (∆∆cq) and normalized versus the actin gene. For microscopic fluorescent observations, the *BbPlin1::eGFP* strain was inoculated into various media and used to infect *G. mellonella* larvae. Fluorescent signals were detected in different cell types and in different developmental stages, as well as during the infection process using a Leica Sp8 microscope. Fungal cells including blastospores, conidia, and hyphae were collected and fixed in 3% formaldehyde for 2–3 h at 4 °C. Formaldehyde–fixed fungal cells were collected by centrifugation and then stained with 500 μL of Nile red (0.5 g/mL) in the dark for 8 min and then washed twice with 1 mL of phosphate–buffered saline (PBS, pH = 7.0). Nile red is a well–characterized fluorescent dye used for lipid detection and the red fluorescent signal of Nile red–stained fungal cells was observed using a confocal microscope.

### 2.4. Lipid Profiles Analysis

Conidia (10^8^ cells) were treated with 280 μL of methanol: water (2:5) in an Eppendorf tube, after which 400 μL of methyl–tert–butyl ether (MTBE) and the sample were mixed by vortexing for 30 s and then homogenized using a probe homogenizer at 45 Hz for 4 min, followed by sonication for 5 min in an ice–water bath. Homogenization and sonication cycles were repeated 4 times. Samples were then incubated at −40 °C for 1 h and then centrifuged at 10,000 rpm for 15 min at 4 °C. A volume of 300 μL of the resultant supernatant was transferred to a fresh tube and dried in a vacuum concentrator at 37 °C. Dried samples were reconstituted in 100 μL of 50% methanol in dichloromethane by sonication on ice for 10 min. The sample was then centrifuged at 13,000 rpm for 15 min at 4 °C, and 75 μL of supernatant was transferred to a fresh glass vial for LC/MS analysis. A quality control (QC) sample was prepared by mixing an equal aliquot of the supernatants from all of the samples. Lipid analyses and separation were carried out using a 1290 Infinity series UHPLC System (Agilent Technologies), equipped with a Kinetex C18 column (2.1 × 100 mm, 1.7 μm), and a triple TOF mass spectrometer was used to acquire MS/MS spectra on an information–dependent basis (IDA) during the LC/MS experiment. In this mode, the acquisition software (Analyst TF 1.7, AB Sciex, MA, USA) was used to continuously evaluate the full scan survey MS data during collection and to trigger the acquisition of MS/MS spectra depending on preselected criteria. An in–house program, Lipid Analyzer, was developed using R for automatic data analysis. The CentWave algorithm in XCMS was used for peak detection.

### 2.5. Effect of BbPlin1 Gene Deletion on B. bassiana Growth

Fungal growth on agar media was examined as follows: two glass slides (25 mm × 75 mm) were separated by three–layer filter paper on both ends and fixed in place using two clips. Agar media (300 μL, CZA or PDA) was added into the space between the two slides. A volume of 20 μL of conidial suspension (10^7^ conidia/mL) was inoculated on the media, the device was placed in a Petri dish with wet filter paper to maintain high humidity, and it was incubated at 26 °C for 3 d after which fungal growth was examined microscopically using Imager.Z2 (ZEISS).

### 2.6. Phenotypic Characterizations and Insect Bioassays

*B. bassiana* strains (2 μL of conidia suspension of 1 × 10^7^ conidia/mL) were spot–inoculated on CZA plates amended with Congo red (250 μg/mL), NaCl (0.7 M), sorbitol (1.2 M), H_2_O_2_ (3 mM), or methylnaphthoquinone (MND, 30 μM). Radial growth was measured after 7 d of incubation at 26 °C.

Insect bioassays were performed using *G. mellonella* larvae. Briefly, conidial suspensions (1 × 10^7^ conidia/mL) were prepared in 0.05% Tween–80 and 30 4th–instar *G. mellonella* larvae were topically inoculated by dipping into the conidial suspension for 5–10 s, after which larvae were blotted on filter paper and transferred to a new Petri dish (150 mm). Controls were treated with sterilized 0.05% Tween–80. All experiments were repeated three times with independent batches of conidia and insects. The number of dead insects was recorded daily, and the median lethal time to kill 50% of hosts (LT_50_) was calculated by probit analysis.

### 2.7. Transcriptome Analysis of Mutant Strains

For transcriptome analysis, the *B. bassiana* wild–type strain and *BbPlin1* deletion mutant were cultured in PDB (50 mL) for 3 d. Total RNA was extracted using TRIzol^®^ following the manufacturer’s instructions (Invotrogen, CA, USA). Genomic DNA was removed with DNase I (Takara) treatment. A mass of 1 µg of total RNA was used for RNA–seq transcriptome library preparation using the TruSeq^TM^ RNA sample preparation Kit (Illumina, San Diego, CA, USA). Libraries (cDNA target fragments of 300 bp) were selected on 2% low–range ultra–agarose followed by PCR amplification (Phusion DNA polymerase, NEB, MA, USA) for 15 cycles. Paired–end RNA–seq libraries were sequenced using an Illumina HiSeq xten/NovaSeq 6000 sequencer (2 × 150 bp read length). Over 50 million reads were obtained for each sample. Differential expression analysis and functional enrichment were performed following the standard protocols by Meiji Biological Company (Shanghai, China).

## 3. Results

### 3.1. Sequence Analysis, Expression Patterns, and Cellular Localization of BbPlin1

A perilipin homolog was identified in the *B. bassiana* genome dataset (NCBI EJP62332) and named *BbPlin1*. The genomic sequence of the *BbPlin1* gene was 624 bp in length with one intron (57 bp) and encoded for a protein of 188 amino acids (20.6 kDa). No signal peptide and/or transmembrane regions were found in the predicted amino acid sequence. Phylogenetic analyses revealed *BbPlin1* to exhibit an 89–98% similarity to homologous proteins from *B. brongniatii* and *Cordyceps* spp., and an 66.1% similarity to the *M. anisopliae* MaMPL1 protein. Phosphorylation of perilipin allows for lipase access to LDs for turnover and TG hydrolysis, with protein modification prediction analyses indicating the presence of 8 putative phosphorylation sites, namely at amino acid residues: T^35^, T^39^, S^49^, S^55^, S^76^, S^84^, Y^108^, and S^137^ (threshold >0.8).

Real–time PCR and GFP fusion construction were used to detect the gene expression and protein expression/localization of BbPlin1 during fungal growth on standard media. *BbPlin1* transcript accumulation showed nutrient dependence, with the highest expression seen on agar plates containing a low level of nutrient (minimal media + sucrose, CZA) followed by growth on PDA, and then on the nutrient–rich SDA media (Figure 1). *BbPlin1* expression was lower (relative to agar plates) in liquid media, but followed the same trend, i.e., CZB > PDB > SDB. Visualization of the expression of a *BbPlin1*::*eGFP* fusion protein under control of its native promoter was used to investigate the subcellular localization and protein levels of *BbPlin1* from cells derived from liquid media (Figure 2) as well as from agar plates (Appendix A). In all cells examined, a punctate pattern of GFP (*BbPlin1*::*eGFP* fusion protein) fluorescent signals was clearly observed that co–localized with Nile Red staining of lipid droplets (LDs) under all conditions. *BbPlin1*::*eGFP* expression signals (and Nile Red staining) were similar in cells derived from CZB and PDB grown for 10 h, with noticeably lower signals seen in 10 h SDB cells. However, by 18 h of growth, only strong signals were seen for CZB cells, with little to no signal (or LD staining) seen for PDB and SDB cells (Figure 2A). *B. bassiana* is known to be able to grow on fatty acid constituents found in insects including oleic acid [15], and the isolation and visualization of blastospores from CZB and CZB amended with 0.25% oleic acid revealed a significant increase in *BbPlin1::eGFP* and Nile Red (LD) staining in the latter cells (Figure 2B). Similarly, cells showed high *BbPlin1*::*eGFP* expression signals during growth on CZA (10 and 16 h), with significantly lower signals seen in cells on PDA and SDA media (Appendix A). In order to examine the effects of starvation on *BbPlin1* expression and LD formation, cells were initially grown in CZB and then transferred to basal minimal salts media (no carbon source, Appendix A). These data showed a rapid loss of *BbPlin1*::*eGFP* fluorescent expression signals (at 6 h post–transfer) that preceded the loss of LDs (as seen by Nile Red staining), where in the case of the latter, reduced (but still visible) Nile Red LD staining was seen 18 h post–transfer, although no *BbPlin1*::*eGFP* expression signals were seen.

### 3.2. Generation of ΔBbPlin1 Mutants and Phenotypic Effects on Hyphal Growth and Lipid Droplet Accumulation

Targeted gene knockout and complemented strains of *BbPlin1* were constructed as detailed in the Methods section (Appendix A). After transformation and screening, three Δ*BbPlin1* deletion mutants were obtained and verified by PCR. The complementation strain was obtained by introducing the entire *BbPlin1* gene with its native promoter into the Δ*BbPlin1* deletion strain. Gene expression in all mutants was further confirmed by reverse–transcription PCR showing complete loss of transcription in Δ*BbPlin1* mutants (Appendix A).

No obvious differences in terms of vegetative growth or colony morphology were seen between the Δ*BbPlin1* and wild–type strains when cultured on various media including 0.5 × SDA, PDA, and CZA over an 8 d time course, or in media supplemented with different stress–causing agents including (i) oxidative: H_2_O_2_ (3 mM) and MND (30 μM), (ii) osmotic: NaCl (0.7 M) and sorbitol (1.2 M), and (iii) cell wall perturbing: Congo Red (250 μg/mL) conditions (Appendix A). The deletion of *BbPlin1* did not affect conidial germination, and GT_50_ values for all strains = 12.0 − 12.5 ± 0.4 h. However, the lipid droplet (LD) staining using Nile Red of cells grown for 20 h in CBZ to allow for full induction time revealed a decreased LD content in Δ*BbPlin1* hyphae (30–40% decrease as compared to the wild–type strain, *p* < 0.01, Figure 3A). The Δ*BbPlin1* mutant also showed accelerated hyphal growth. After culturing on CZA for 16 h, about 50% of Δ*BbPlin1* hyphae exceeded 10 nm in length with an overall distribution of longer hyphae, which were significantly (*p* < 0.01) longer than those seen for the wild–type and complemented strains (Figure 3B). Microscopic (including stereoscopic) observations of hyphae grown on solid media, including CZA, PDA, and 0.5 × SDAY, confirmed that the Δ*BbPlin1* strain produced a greater abundance and longer aerial hyphae as compared to wild–type and complemented strains (Figure 3C,D).

### 3.3. Loss of BbPlin1 Has a Minor Effect on Fungal Virulence

To determine the effect of *BbPlin1* on fungal virulence, conidia harvested from PDA plates were inoculated onto *G. mellonella* larvae. Compared to the wild–type strain, deletion of *BbPlin1* showed a small but significant decrease (~13%, *p* < 0.01) in fungal virulence, shifting the mean lethal time to 50% mortality (LT_50_) from 109.2 ± 1.5 h for the wild–type and complemented strains to 123.6 ± 4.2 h for the Δ*BbPlin1* mutant (Figure 4A). Although the deletion of *BbPlin1* had only a slight effect on fungal virulence, a faster growth of mutant hyphae was observed on cadavers, and the conidial yield was significantly increased as compared to wild–type–killed cadavers (*p* < 0.01, Figure 4B,C). In addition, production of the *BbPlin1* protein during the infection process was analyzed by use of the *BbPlin1*::*eGFP* fusion expression strain (*BbPlin1::eGFP*). Conidia from the *BbPlin1::eGFP* strain were injected into the hemocoel of *G. mellonella* larvae and *BbPlin1*::*eGFP* signals in the fungal cells monitored via fluorescent microscopy over a time course of the infection process. The initial production of hyphal bodies within the insect hemocoel showed weak fluorescent signals in these cells (36–60 h, Figure 5, note host death occurs ~60 h). After host death (>96 h), significantly increasing GFP (and co–localizing Nile Red) staining could be seen in hyphae growing from the hyphal bodies up to 144 h initial post–inoculation (~70–90 h post–death), with hyphae almost saturated with LDs. However, as the fungus grew out from cadavers (144 h and onwards), *BbPlin1*::*eGFP* signals and concomitant Nile Red–stained LDs rapidly decreased, until little to no staining was visible in hyphae, although LDs could be detected in the conidia produced on the surface of cadavers. Fluorescent microscopic observations of cross–sections of infected cadavers confirmed intense but almost exclusive fluorescent signals within the cadaver but none from fungal cells growing on the outside surface of the cadaver. A control strain constitutively expressing *eGFP* showed fluorescent signals from both internal and external growing fungal cells within/on the insect cadaver (Appendix A). In order to further confirm that *BbPlin1* was essentially only expressed by cells growing within the insect cadaver (within the context of the infection cycle), mycelia from within the cadaver and cuticle surface growing mycelia were separated, and the total RNA was extracted from these two samples. As the *BbCal1* caleosin is also an important LD–associated protein, we examined its expression as well as *BbPlin1* on the surface of insect cadavers, conditions where *BbPlin1* expression is low, in order to see whether there may be any compensatory effects on gene expression (Appendix A). These data show that both *BbPlin1* and *BbCal1* were highly expressed in cells isolated from within the cadaver, but almost completely lacking in the cadaver surface growing fungal hyphae. In order to complete the analyses of *B. bassiana* cells from cadavers, conidia produced on the cadavers were examined for *BbPlin1*::*eGFP* expression using the respective expression strain, via fluorescent microscopy (Appendix A). These results showed that conidia immediately harvested from cadavers expressed little to no *BbPlin1*, but as the cells swelled (in PDA media, 9 h), *BbPlin1* expression was induced, accumulating in the conidial base as the cell germinated (11 h post–inoculation in media), and then distributed into the punctate patterns of LDs in germlings (14 h post–inoculation in media).

### 3.4. Compensatory Expression of Bbcal1 and Transcriptomic Analyses of the ΔbbPlin1 Mutant

Perilipins and caleosins are the two major LD–associated structural proteins found in fungi, and as both *Plin1* and *Cal1* act to control LD formation and turnover, we thought it important to determine whether the deletion of one would affect the expression of the other, potentially leading either to compensatory or even additive effects. Gene expression analyses revealed that *Bbcal1* expression was increased ~5–fold in Δ*bbPlin1* cells grown in low–nutrient media (CZB), but was unchanged in richer PDB and SDB media (Figure 6A). However, the expression of *BbPlin1* in a previously constructed Δ*bbcal1* strain indicated little to no change in *BbPlin1* expression under the same conditions (Figure 6B). To further evaluate consequences of loss (targeted gene deletion) of *Bbcal1* on *BbPlin1* expression, the *BbPlin1::eGFP* fusion construct was introduced into the Δ*bbcal1* mutant. Fluorescent microscopic observations further confirmed that deletion of *Bbcal1* does not appear to affect the expression and localization of *BbPlin1*::*eGFP* in LDs (Figure 6C).

To explore the effect loss that *BbPlin1* has on global gene expression, a transcriptomic analysis was performed comparing wild–type and Δ*bbPlin1* cells grown in PDB for 3 d, as detailed in the Methods section. PD medium was used as the basis for comparison as this medium represents the conditions used for the insect bioassays, the analyses of the lipid profiles given below, and would allow for comparisons to transcriptomic data from other *B. bassiana* mutants that have been most commonly performed with cells grown in PD. Compared to the wild–type strain, deletion of *BbPlin1* resulted in 1353 differentially expressed genes (DEGs, log2F > 1), with 662 upregulated genes and 691 downregulated genes. The functional category of differentially expressed genes indicated that upregulated genes are mainly related to energy production and conversion, carbohydrate transport and metabolism, and lipid transport and metabolism, and downregulated genes are related to cell cycle and signal transduction mechanisms (Appendix A). In terms of genes involved in lipid pathways, including fatty acid degradation, fatty acid biosynthesis, and glycolipid metabolism, more upregulated genes were observed in the *BbPlin1* mutant as compared to the wild–type strain (Up/Down, 19/7) (Appendix A). RT–PCR analyses confirmed the expression of several lipid metabolism–related genes, including triacylglycerol lipase (*TGL–3*), Enoyl–CoA hydratase (*Enoyl–CoA*), and diacylglycerol o–acyltransferase (*Dga–1*), significantly regulated in the *BbPlin1* deletion mutant (Figure 7).

### 3.5. Loss of BbPlin1 Alters Cellular Lipid Profiles

Lipid profiles were examined by comparing the Δ*BbPlin1* strain to the wild–type strain in conidia harvested from PDA and PDA + 0.05% oleic acid, where relative lipid content levels were analyzed by percentages calculated by the area–normalized method, as detailed in the Methods section. In total, 12 classes of lipid components were detected, with the most abundant species enriched in phosphatidylcholine (PC), triglyceride (TG), phosphatidylethanolamine (PE), and ceramide (CER), which contained 47, 37, 26, and 12 subspecies, respectively. The top three classes found in the wild–type strain were TG, PE, and PC. The Δ*BbPlin1* mutant conidia showed a significant (2–3 folds, *p* < 0.01) reduction in TG, diglyceride (DG), and PE levels as compared to the wild–type strain. For both the wild–type and Δ*BbPLin1* strains, growth in the presence of oleic acid resulted in the accumulation of TG in conidia, but significantly reduced PE content. However, even under these conditions, the concentrations of most species of TG, DG, and PC were reduced in the Δ*BbPlin1* mutant as compared to the wild–type parent (Figure 8).

## 4. Discussion

Lipid storage is one of the main functions of lipid bodies, and their formation in filamentous fungi reflects lipid storage and turnover requirements in meeting various fungal lifestyles [2,30,31]. LD–associated proteins regulate the formation and turnover of LDs, and play important roles in fungal growth, development, and, where applicable, virulence. Our results indicate that the *B. bassiana* perilipin (*BbPlin1*) is an LD–associated protein that affects but is not essential for fungal growth and infection toward insect hosts. *BbPlin1* was localized to LDs, regulating the formation of LDs and their turnover. Both transcript and protein expression patterns of *BbPlin1* tightly correlated with LD content, with both responsive to nutrient availability in which low–nutrient conditions appeared to favor perilipin and LD expression/accumulation and high–nutrient conditions repressed expression/LD formation. These results also suggest that under nutrient limiting (i.e., in CZB media), *B. bassiana* produces and stores lipids by de novo lipid synthesis as no exogenous TG sources are available. Nutrient availability is known to be an important factor affecting lipid production in fungi [32,33]. The plant fungal pathogen, *Ustilago maydis*, accumulates LDs when cultured in the absence of a nitrogen source, and the lipolytic yeast, *Yarrowia lipolytica*, has been shown to accumulate significantly greater amounts of TAGs when cultured in minimal media as compared to growth in rich media [33,34]. It should be noted that the in vitro nutrient “low” conditions of CZB still include appreciable amounts of carbon (0.5% sucrose) and that “starvation” or very low nutrient conditions such basic salts media without a carbon source or the surface of the insect cuticle cadaver surface results in the very low expression of *BbPlin1*. With respect to the surface of the insect cadaver, the low expression of the perilipin (and the absence of LDs) may reflect the direction of fungal resources to conidial production without the benefit/ability to store excess carbon in LDs, as is seen during growth in CZB media.

Our data indicate that *B. bassiana* conidia produced in vitro (i.e., on mycological media, e.g., CZA, PDA) contain an initial store of LDs that may provide energy for germination and germling growth. Cells then accumulate LDs in growing hyphae and mycelia, with less seen in growth tips. During infection of insect hosts, entomopathogenic fungi are known to be able to detoxify and/or assimilate specific insect cuticular compounds, including lipids that impact virulence [34]. In *M. anisopliae*, the mobilization of lipid storage via the activity of a perilipin has been shown to play an important role in establishing the turgor pressure in the appressorium [12]. However, *B. bassiana* does not produce similar appressorial structures and the need for lipid mobilization during cuticle penetration is not clear [18,19]. Our data do indicate that *BbPlin1* makes a small contribution to virulence, with only a ~13% decrease in virulence seen for the Δ*BbPlin1* mutant. This effect is likely to have low biological significance and differs from the effect seen in *Metarhizium* [12]. Our data indicate, however, that the perilipin in *B. bassiana* may be activated during the within–host (in vivo) utilization of nutrients from the hemocoel and other internal insect tissues, particularly at the post–death phase of the infection process. Recent studies have shown that post–morbidity events can have direct impacts on fungal fitness via a number of different mechanisms, with a major one being the ability of the fungus to compete/antagonize bacterial competitors [35]. Specifically, *B. bassiana*, the pigmented secondary metabolite, oosporein, is expressed and acts as an antibiotic, primarily after the host has died in order to inhibit the growth of competing bacteria [35]. In this context, although lipid storage and turnover are known to be important for the interactions between entomopathogenic fungi and insect hosts, little is known about lipid formation during the infection process within the host or during the later stages of fungal development on hosts, i.e., after host death. Using a BbPlin1::eGFP–expressing fusion strain, coupled to Nile Red–staining of LDs, our data show that both perilipin and LD formation is low in the free–floating yeast–like hyphal bodies produced in the insect hemocoel by the fungus after cuticle penetration, which represents an intermediate stage of infection, i.e., after the cuticle has been penetrated but before death of the host. However, perilipin and concomitant LD accumulation to high levels occurs in fungal cells growing inside the insect post–host death. It is noteworthy that high perilipin/LD accumulation was seen in cells within the dead host but not on hyphae and mycelia that had reverse–penetrated the cuticle from the inside out to grow on the cadaver. Thus, in marked contrast to in vitro growth in which high *BbPlin1*/LD levels are seen in nutrient–poor media but low levels are seen in richer media, our data show that utilization of the relatively nutrient–rich resources of the insect body, which includes lipid stores, promotes perilipin/LD accumulation, but that once on the nutrient–poor surface of the insect cadaver, where the fungus must still form conidia in order to complete its life cycle, the lipid (LD) nutrient stores that have accumulated from the utilization of internal host nutrients are likely used up for the process of conidiation (on the cadaver). This is consistent with the finding that *BbPlin1* expression is very low in conidia produced on cadaver; however, as these conidia germinate in a suitable nutrient environment, perilipin and LDs once again begin to accumulate.

Spore germination and hyphal growth of *B. bassiana* have been shown to be affected by insect lipids [36,37], and more recently, a lipid–binding aegerolysin protein has been characterized from *B. bassiana*, although the role of the latter protein in fungal virulence and insect lipid utilization was not directly examined [38]. The *B. bassiana* perilipin did not seem to significantly impact stress tolerances, as the Δ*BbPlin1* mutant showed no obvious changes in terms of vegetative growth in standard media or media containing various stress–causing agents, including H_2_O_2_ (oxidative stress), Congo red (cell wall–perturbing stress), NaCl, and sorbitol (the latter two, osmotic stress). Deletion of *BbPlin1* also did not affect rates of conidial germination. However, intriguingly, a more rapid hyphal growth was observed in the mutants as compared to the wild–type strain, with noticeably longer aerial hyphae apparent for the former strain. As expected, Δ*BbPlin1* mutants showed reduced LD accumulation, and it is possible that the inability of the cells to store lipids results in their (more rapid) utilization, which can account for the greater hyphal extensions seen in the mutant. This is consistent with the observation that, in addition to their involvement in LD formation, perilipins regulate LD turnover by blocking access of metabolic enzymes LDs [7]. Thus, knockout of the mouse perilipin resulted in an ~30% reduction of adipose tissue compared to that in wild–type animals [39], presumably as the loss of LD protection from degradation by the perilipin increased basal lipolysis activity in the isolated adipocytes [39]. Consistent with this, we observed that the deletion of *BbPlin1* resulted in reduced levels of triglyceride, diglyceride, and phosphatidylethanolamine, and concomitant increases in the expression of lipid metabolic–related genes, including triacylglycerol lipase, enoyl–CoA hydratase, and diacylglycerol o–acyltransferase, were noted, thus contributing to a faster turnover of lipid contents in the perilipin mutant, impacting the hyphal growth rate.

Perilipins function in concert with another major LD–associated protein, namely caleosins. Our previous study indicated that the *B. bassiana* caleosin (*Bbcal1*) is also involved in LD formation [28]. *Bbcal1* and *BbPlin1* exhibit similar expression patterns under various culture conditions and they both contribute to LD homeostasis. Indeed, the phenotypes of the Δ*BbPlin1* and (previously reported) Δ*BbCal1* mutants share a number of similarities including reduced cellular lipid contents and increased hyphal growth. Our data show that the deletion of *BbPlin1* resulted in increased *BbCal1* expression, suggesting cellular attempts at compensating for loss of the perilipin. However, in the Δ*BbCal1* mutant, no similar increased *BbPlin1* expression was seen. A number of additional lipid pathway proteins have been characterized in entomopathogenic fungi, including a peroxisomal sterol carrier protein (*BbScp2*) that was shown to be involved in insect lipid trafficking [40], and a sterylacetyl hydrolase (*BbSay1*) that contributes to oleic acid production and lipid homeostasis, with the Δ*BbSay1* mutant showing elevated LD accumulation resulting in impaired conidial production and germination, as well as decreased virulence [41]. Furthermore, a secretory phospholipase (PLA2) was found to be exclusively expressed in B. bassiana insect–derived hyphal bodies, and whose regulation was mediated by nutrient availability in a pattern similar to the *B. bassiana* perilipin [42]. Constitutive expression of *BbPLA2* increased LD content, enhanced the ability of *B. bassiana* to utilize (insect) lipids, and promoted virulence. It is likely that the perilipin works in concert with the phospholipase and other factors to maintain lipid homeostasis. These findings together with our results highlight the importance of LD and LD processes in the physiology and virulence of entomopathogenic fungi, suggesting an important co–evolution between lipid assimilation and virulence in targeting the rich lipid substrates present on the insect host.

## 5. Conclusions

Lipid storage and turnover are important in entomopathogenic fungi to meet various lifestyles. LD–associated proteins regulate the formation and turnover of LDs, and play important roles in fungal growth, development, and virulence. However, there are only a few LD–associated proteins that have been characterized in entomopathogenic fungi. In the present study, we characterized a perilipin in *B. bassiana* (*BbPlin1*), which localized to LDs and regulated the formation of LDs. *BbPlin1* expression is affected by nutrient availability, and low–nutrient conditions appeared to favor perilipin expression. Deletion of *BbPlin1* resulted in the reduction in cellular LD content, promotion of aerial hyphal growth, and decreased virulence. However, further studies are required to unveil the relationship between BbPlin1 and other LD–associated proteins in regulating lipid metabolism. These data expand our understanding of lipid processes in fungal growth and development, and during host–pathogen interactions.

## Figures and Tables

**Figure 1 jof-08-00634-f001:**
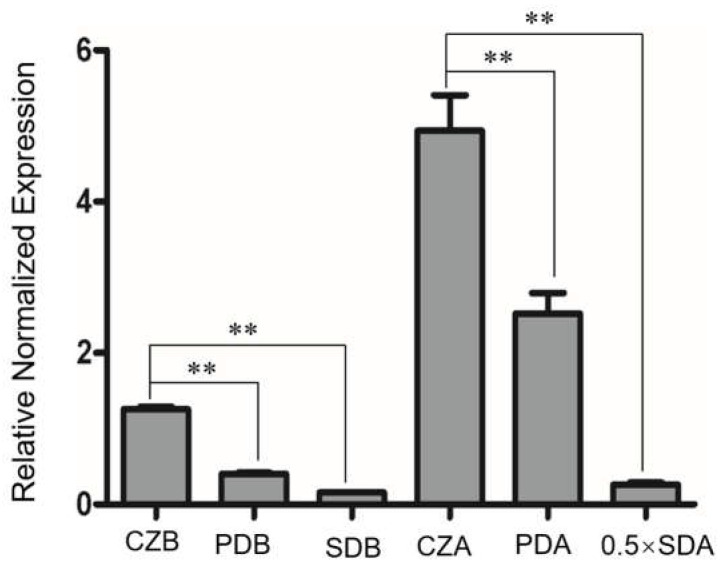
Gene expression analysis of *BbPlin1*. Wild–type *B. bassiana* was cultured in liquid media including CZB, PDB, and SDB for 3 d; on solid media, CZA, PDA, and 0.5 × SDA were cultured for 7 d; and total RNA was extracted from fungal cells for use in RT–PCR experiments as detailed in the Methods section. All experiments were performed using three technical replicates and the entire experiment was performed three times. Data are given as ± SE. Stars represent statistical difference (** *p* < 0.01, *t* test).

**Figure 2 jof-08-00634-f002:**
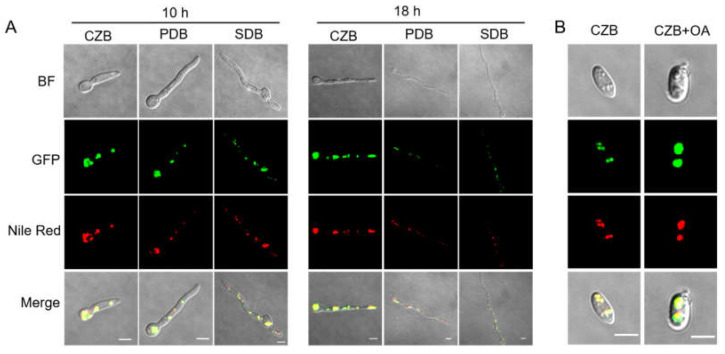
Representative images of *BbPLin1*::eGFP fusion protein expression and cellular localization. (**A**) *B. bassiana* harboring the *PBbPlin1::eGFP* construct was cultured in indicated media (CZB/PDB/SDB) for 10 and 18 h, after which fungal cells were counter–stained with Nile Red. (**B**) Effects of oleic acid on lipid droplet (LD) formation. *B. bassiana PBbPlin1::eGFP* conidia were cultured in CZB and CZB + 0.25% oleic acid for 4 d before visualization of cells using fluorescent confocal microscopy. BF: bright field. Size bar = 5 μm.

**Figure 3 jof-08-00634-f003:**
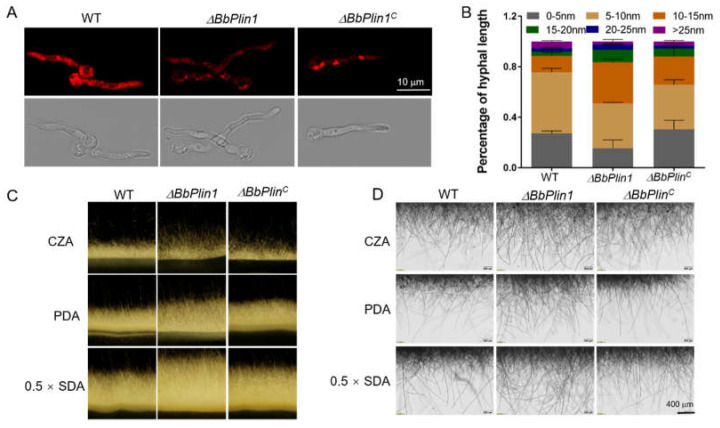
*BbPlin1* affects LD formation and hyphal growth. (**A**) The *B. bassiana* wild–type and Δ*BbPlin1* strains were inoculated into CZB and cultured for 20 h after which fungal cells were collected by centrifugation, stained with Nile Red, and visualized by fluorescent microscopy, as described in Materials and Methods. (**B**) Quantitative measurement of fungal hyphal lengths. Fungal conidia were cultured on CZA for 16 h and the hyphal length was measured microscopically via ImageJ analyses. (**C**,**D**) Effects of *BbPlin1* on aerial hyphae growth. Conidial suspensions of wild–type and mutant strains were inoculated onto CZA, PDA, and 0.5 × SDAY media placed into the space between two glass slides, as described in the Materials and Methods. Samples were cultured at 26 °C for 3 d and observed by a fluorescent macroscopic stereomicroscope and automatic inverted DIC microscopy.

**Figure 4 jof-08-00634-f004:**
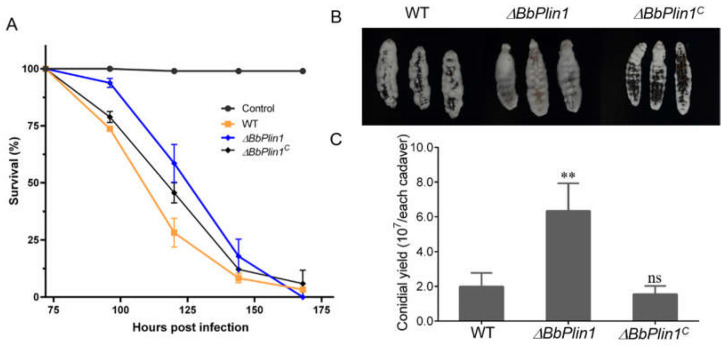
Insect bioassays: virulence analyses of wild–type *B. bassiana* and Δ*BbPlin1*. (**A**) Insect survival curves and treatment with conidial suspensions (1 × 10^7^ conidia/mL) of indicated strains topically inoculated onto *G. mellonella* larvae, with survival recorded every 24 h. (**B**) Representative images of fungal growth on cadavers 5 d post–death. (**C**) Quantification of fungal conidial production on cadavers 20 d post–death. Fungal conidia were harvested from 2–week–old PDA plates of *B. bassiana* wild–type (WT) strain and mutants (Δ*BbPlin1*, Δ*bbPlin1^C^*) grown at 26 °C. All experiments were performed using three technical replicates and the entire experiment was performed three times. Data are given as ±SE. Stars represent statistical difference (** *p* < 0.01; ns, no significant, *t* test).

**Figure 5 jof-08-00634-f005:**
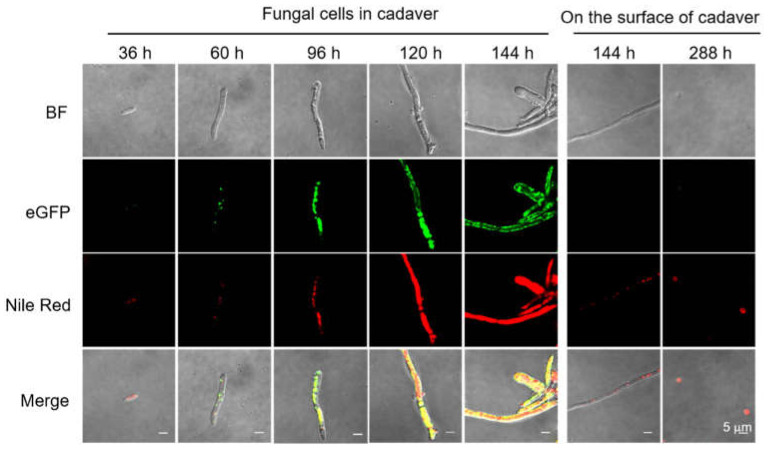
Representative images of *BbPlin1*::*eGFP* expression during the infection process using *G. mellonella* larvae as the host. Conidia from *B. bassiana* harboring the *PbbPlin1*::*eGFP* construct (5 µL/insect, 1 × 10^6^ conidia/mL) was injected into the haemocoel of *G. mellonella* larvae. Images of fungal cells isolated at indicated time points and that include (i) fungal hyphal bodies in hemolymph (36 h), (ii) fungal hyphae formed post–host death inside the host (60–144 h), (iii) fungal hyphae formed on the surface of the cadaver (>144 h), and (iv) fungal conidia produced on the surface of the cadaver (288 h) were collected and counter–stained with Nile Red. Fungal conidia were harvested from *BbPlin1::eGFP* grown on PDA for 2 weeks at 26 °C. eGFP and Nile Red signals were visualized using confocal fluorescent microscopy as described in the Methods section. BF, bright field. Bar, 5 μm.

**Figure 6 jof-08-00634-f006:**
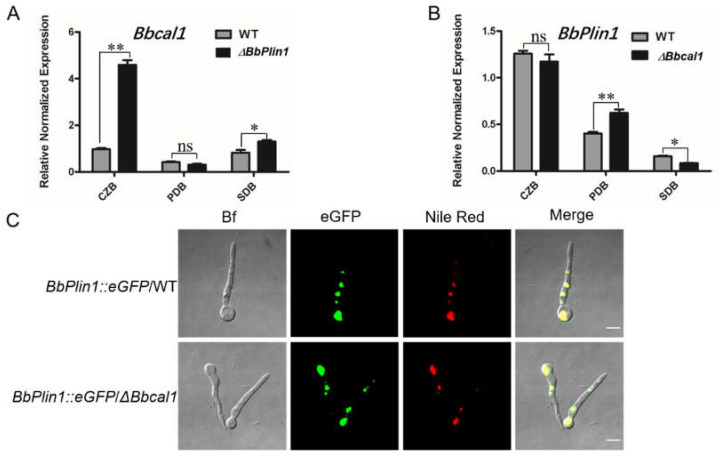
Effects of *BbCal1* and *BbPlin1* targeted gene deletion on gene expression of *BbPlin1* and *BbCal1*, respectively. (**A**) Gene expression analysis of *BbCal1* in *B. bassiana* wild–type strain and Δ*bbPlin1* deletion mutant. Fungal cells were inoculated into CZB, PDB, and SDB for 3 d and total RNA was extracted for use in RT–PCR experiments. (**B**) *BbPlin1* expression in *B. bassiana* wild–type strain and Δ*bbcal1* deletion mutant. (**C**) Representative images of eGFP fluorescence signals of *B. bassiana* Δ*bbCal1* harboring the *PbbPlin1::eGFP* construct. Fungal conidia were harvested from indicated *B. bassiana* strains grown on PDA for 2 weeks at 26 °C. All experiments were performed using three technical replicates and the entire experiment was performed three times. Data are given as ± SE. Stars represent statistical difference (* *p* < 0.05; ** *p* < 0.01; ns, no significant, *t* test).

**Figure 7 jof-08-00634-f007:**
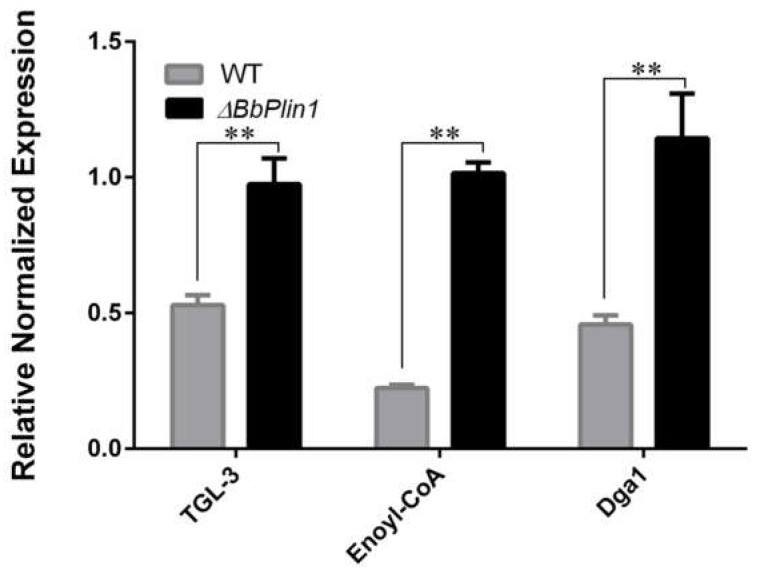
Effects of *BbPlin1* on the expression of lipid metabolism–related genes. After culture, *B. bassiana* wild–type and Δ*bbPlin1* deletion strains were grown in PDB for 3 d, and the total RNA was extracted and used for RT–PCR analyses of the expression of the triacylglycerol lipase (*TGL–3*), Enoyl–CoA hydratase (Enoyl–CoA), and diacylglycerol o–acyltransferase (*Dga1*) genes. All experiments were performed using three technical replicates and the entire experiment was performed three times. Data are given as ±SE. Stars represent statistical difference (** *p* < 0.01, *t* test).

**Figure 8 jof-08-00634-f008:**
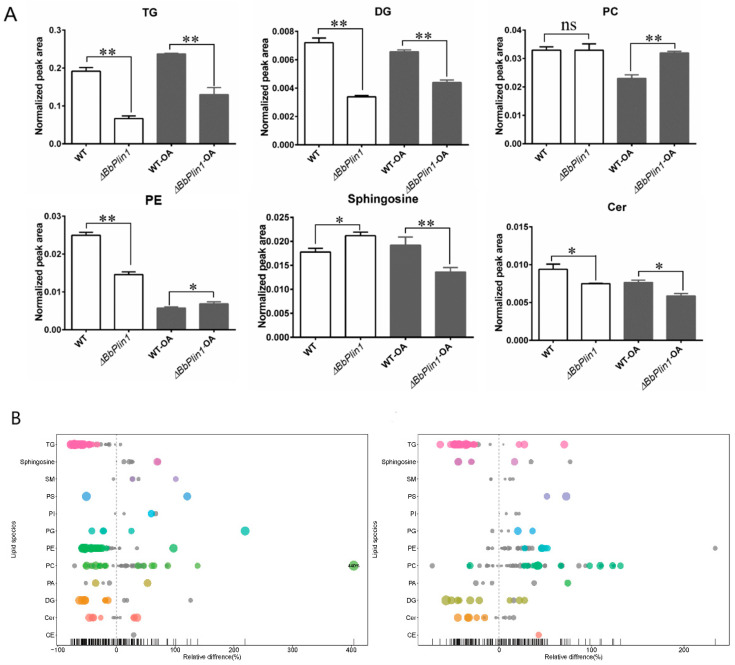
Lipid profiles analyses of *B. bassiana* wild–type strain and Δ*BbPlin1* conidia harvested from PDA and PDA + 0.25% oleic acid (OA). (**A**) Calculated amounts of lipid contents based on normalized peak area. (**B**) Bubble plot of relative percentages of detected lipid species in the Δ*bbPlin1* mutant as compared to the wild–type strain grown with and without oleic acid. Each circle represents a different lipid species. Colored circles indicate significance with *p*–values of differences in the content of the one specific lipid species <0.05, with larger circle sizes indicating even smaller *p*–values. Gray spots indicate lack of significance, where the *p*–value is 0.5. The X–axis indicates relative percentages, with X = 0 indicating equivalent amounts of a lipid species in both wild–type and mutant cells. For any specific lipid component, X > 0 indicates higher levels in the Δ*BbPlin1* mutant than in the wild–type strain, whereas X < 0 indicates lower levels of indicated lipid species in the mutant as compared to the wild–type strain. All experiments were performed using three technical replicates and the entire experiment was performed three times. Data are given as ±SE. Stars represent statistical difference (* *p* < 0.05; ** *p* < 0.01; ns, no significant, *t* test).

## Data Availability

All materials are available by the corresponding author.

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
