# Peer review of "A Perilipin Affects Lipid Droplet Homeostasis and Aerial Hyphal Growth, but Has Only Small Effects on Virulence in the Insect Pathogenic Fungus Beauveria bassiana"

_jof, 2022, doi:10.3390/jof8060634_

Round 1

Reviewer 1 Report

This is an interesting study on lipid storage and turnover in entomopathogenic fungi. The authors targeted a perilipin from B. bassiana to understand its function, and their results nicely complement the knowledge of this field together with previous results about a perilipin in the related fungus M. anisopliae, and regarding other lipid-related proteins such as caleosin, sterol carrier protein, sterylacety hydrolase, and phospholipase recently characterized in B. bassiana. The methodological approach is scientifically sound, including reverese genetics, transcriptomics, and a bounded lipidomics. Some points need to be improved/clarified:

Title: it states that perilipin contributes to hyphal growth, but then the manuscript (specifically the paragraph ending in line 278) states that the perilipin mutant produces greater abundance and longer aerial hyphae than wild type and complemented strains. This sounds contradictory, please clarify.

Introduction: please cite the sentences in lines 42-46 and 76-78. This last one in very interesting to help understand the different strategies for insect infection between both fungi. On the other hand, the sentence in lines 46-47 about metabolic diseases is not a fungus-related one, it seems out of context and thus should be deleted.

Methodology: Regarding gene expression, it is strange the way that relative expression is shown. It is more common to refer as ΔΔCq. I am not expert, but I understand that besides normalization with a housekeeping gene, the expression in each medium should be relatively compared with a control medium. Please explain and provide bibliography showing ΔCq is commonly used. If so, please provide in the ms the equation to guide other researchers how to calculate it.

Regarding Nile red, please clarify that it is commonly used as specific dye for lipid droplets, it should be useful for understanding of readers not specialized in biochemistry.

Results: Clarify in line 202 (or in another place that authors consider better) that phosphorylation of perilipin is key to play its function regarding allowing lipases to entering LD and starting TG hydrolysis.

Figure 1: Is it showing results from solid or liquid media? The caption mentions both but there is only one result (unless they are normalized to each other). Please clarify.

In all bar charts, please highlight significant differences with asterisks or different letters.

In line 388 it reads that for both strains there is accumulation of TG in conidia, but in next line reads that the concentrations of most species of TG were reduced. Thus, is any species of TG accumulating? What species would it be? Pink bubbles is Fig. 8B do not seem to be different in both panels.

Discussion: The authors state that there is perilipin/LD accumulation in insect body (cadaver), but that once on the nutrient poor surface of the insect cadaver both significantly drop. However, the authors also found that in a synthetic nutrient poor media (Czapek-Dox) both perilipin and LD accumulate. How do the authors explain this difference between two poor media (i.e. synthetic and natural)?

Minor comments:

Check for a space between “Δ” and “BbPlin1” throughout the manuscript and correct it. Also, check typography since “Δ” was changed in some places and it is showed with a weird symbol in front.

Italicize scientific names in Discussion section.

Delete repeated word (of) in line 440.

Line 447: little is known

Author Response

Dear editor and reviewers  
    We are grateful to you for the constructive comments on our manuscript. As per the suggestion made by the two reviewers, the title has been modified to “A perilipin affects lipid droplet homeostasis and aerial hyphal growth, but has only small effects on virulence in the insect pathogenic fungus Beauveria bassiana” to better reflect the main aspect of the paper. The manuscript has been revised based on the Reviewers’ suggestions. Introduction, Results, Discussion, and Materials and Methods have been modified/updated in response to specific reviewers’ comments. The changes in the revised version were marked in red. We feel that the quality of our paper has been substantially improved after revision and appreciate your consideration of this work. 

Sincerely yours,

Yanhua Fan

Reviewer 2 Report

The work entitled “A perilipin contributes to lipid droplet homeostasis, aerial hy- 2phal growth and virulence in the insect pathogenic fungus Beauveria bassiana”explore the perilipin function of the entomopathogenic fungus Beauveria bassiana. The assays were well conducted, and appropriated controls were used. In general, the manuscript brings new insights about the role of lipids on the biology and virulence of fungi. The authors approached the perilipin function using a variety of techniques such as, qPCR, in silico data, gene deletion, RNAseq and lipid profile analysis. However, this work has serious problems on conclusions not supported ny the collected data and/or the rational of the conditions used in some assays. 

Major points:

The data did not analyzed the role of the perilipin on lipid metabolism in B. bassiana. Thus, the title must be adjusted accordingly. 

Introduction

Add more details about the caleosin-associated functions in fungi and other organisms.

Lines 98, 99: move to the conclusion section

Methods:

No information on transcriptome analysis was provided. The authors must depict on how RNAseq was conducted.

Results

Lines 198-201: the phylogenetic tree was not explored. It is not necessary to build a phylogenetic tree to obtain homology/similarity values for a specific sequence. The authors should either remove the tree or describe the phylogenetic aspects revealed by the analysis. 

Line 219: replace expression by transcript accumulation

Line 224/225: replace expression pattern by protein levels

Since authors used different media on the LD content and Perilipin level analysis the composition of these media must be shown. 

Lines 232-234: what is the rational for the oleic acid treatment?

Panel 3D is not cited in the text.

Line 259 what is SDAY medium?

Figure 3A: why the LD content was performed in cells cultured in CZB medium at 20h?

Line 280: why infection assays were performed  wth cells cultured in PDA medium that 

Lines 280-287: it seems complemented strain does not recovered virulence at WT levels. The difference of LT50 between deleted and WT strains is very subtle and therefore is not biologically relevant. This is reinforced by the fact that LD and Plin1 accumulated after host death (lines 287-302). 

Line 308: why Cal1 transcripts were measured at this time?

Lines 338-347: the results on the Plin1 deletion on Cal1 transcript accumulation are not informative and have no rational. 

Lines 364 – 377: the transcriptome analysis was performed in PDA cultivated cells, that does not present a higher LD content.

Line 380: again, why PDA medium was used in lipid profile assays?

Line 416: the data do not support this conclusion

Lines 418-420: the data presented in figure S3 contradict this statement, since transference to salt solution decreases LD content. 

Line 428: italicize B. bassiana and in vitro

Lines 428-30: not supported by the data presented.

Lines 437-41: contradictory statements. The very slightly reduced LT50 does not couple with LD like structures in Galleria infection model

Line 447: typo Knowns

Lines 467-68: not informative

Author Response

(The authors gave the same response as above.)

Round 2

Reviewer 1 Report

Authors have responded satisfactorily to all my comments

Author Response

Thanks for the positive comment. 

Reviewer 2 Report

see the comments in the attached file

Author Response

Dear reviewer  
    We are grateful to you for your careful review. As per your suggestion, we added more information in the manuscript. The changes in the revised version were marked in red. We feel that the quality of our paper has been substantially improved after revision and appreciate your consideration of this work. 

Sincerely yours,

Yanhua Fan